

# Effectiveness of newborn infant parasympathetic evaluation (NIPE) index for guiding intraoperative fentanyl administration in children under 2 years: a randomized controlled trial

Darunee Sripadungkul, Sirirat Tribuddharat, Thepakorn Sathitkarnmanee, Pimprapa Muenpirom and Panaratana Ratanasuwan

Department of Anesthesiology, Faculty of Medicine, Khon Kaen University, Khon Kaen, Thailand

## ABSTRACT

**Background:** Assessing pain in infants and neonates is challenging due to their inability to communicate verbally. While validated subjective tools exist, they rely on interpreting the child's behavior, leading to potential inconsistencies and underestimation of pain. Based on heart rate variability, the newborn infant parasympathetic evaluation (NIPE) index offers a more objective approach to pain assessment in children under 2 years. Although promising, research on its effectiveness during surgery under general anesthesia remains limited and inconclusive.

**Objective:** This study compared the effectiveness of NIPE-guided fentanyl administration to traditional vital signs (heart rate and mean arterial pressure) in managing pain during surgery in children under 2 years.

**Methods:** Seventy children undergoing head, neck, or upper extremity surgery were randomized into group N (NIPE) or group C (Control) with 35 participants in each group. Both groups received standardized anesthesia and monitoring, including NIPE. Group N received fentanyl when NIPE scores fell below 50, while group C received fentanyl upon a 20% increase in heart rate or mean arterial pressure (MAP). Postoperative pain was assessed using the Face, Legs, Activity, Cry, and Consolability (FLACC) scores, alongside fentanyl consumption, sedation levels, and potential side effects.

**Results:** Both groups exhibited similar outcomes in terms of postoperative pain scores, fentanyl consumption, sedation levels, and absence of side effects (postoperative respiratory depression or nausea and vomiting). Additionally, intraoperative NIPE scores, heart rate, and MAP were comparable between the groups. There are strong correlations between both groups in terms of NIPE scores ($r = 0.735$, $p < 0.0001$), heart rate ($r = 0.630$, $p < 0.0001$), and MAP ($r = 0.846$, $p < 0.0001$). In both group C and group N, the NIPE scores reveal strong negative correlations with heart rate ($r = -0.610$, $p < 0.0001$, and $r = -0.674$, $p < 0.0001$) and MAP ($r = -0.860$, $p < 0.0001$, and $r = -0.756$, $p < 0.0001$).

**Conclusion:** NIPE-guided intraoperative fentanyl administration was not superior to heart rate/MAP-guided administration, as both achieved similar pain management

Corresponding author
Sirirat Tribuddharat,
sirirat.tribuddharat@gmail.com

outcomes in this study. However, NIPE may offer a more practical and precise approach, as it is an objective tool with a defined threshold. These findings suggest NIPE's promise as a valuable tool for managing pain in children under 2 years undergoing surgery. However, confirmation of its widespread effectiveness requires further research with larger, multicenter studies encompassing procedures with a broader spectrum of pain severity.

# INTRODUCTION

Balanced general anesthesia utilizes a combination of hypnotics, opioids, and muscle relaxants to achieve optimal surgical conditions. Reliable monitoring is essential to maintain these drugs at therapeutic levels and ensure adequate anesthesia depth. The bispectral index (BIS) (*Oliveira, Bernardo & Nunes, 2017*) and minimum alveolar concentration (MAC) (*Aranake, Mashour & Avidan, 2013*) track the hypnotic state for intravenous and inhaled medications, respectively. The train-of-four ratio with a nerve stimulator effectively monitors muscle relaxation (*Murphy, 2018*). However, there's no single, reliable monitor for opioids. Clinicians rely on the drug's pharmacokinetic profile (*Duthie, McLaren & Nimmo, 1986*) and observe indirect signs like sweating, tears, pupillary changes, heart rate variations, and blood pressure fluctuations. Unfortunately, the lack of specific indicators can lead to under- or over-dosing of opioids, potentially causing intraoperative awareness, postoperative pain, respiratory depression, and nausea and vomiting (PONV) (*Upton et al., 2017*).

Accurately assessing pain in neonates and infants is a challenge because they can't communicate their discomfort verbally. While validated subjective tools like the Comfort Behavioral (COMFORT-B) scale, Premature Infant Pain Profile (PIPP), and the Face, Legs, Activity, Cry, and Consolability (FLACC) scale exist, they rely on interpreting the child's behavior. This inherent subjectivity can lead to potential inconsistencies and underestimation of pain (*Ivanic et al., 2023*).

Heart rate variability (HRV) reflects the activity of the autonomic nervous system, offering insights into how the body responds to pain and stress. HRV allows us to measure the balance between the sympathetic (fight-or-flight) and parasympathetic (rest-and-digest) branches of the nervous system by analyzing variations in heartbeats. The Analgesia Nociception Index (ANI™; MDoloris Medical Systems, Loos, France) is a tool derived from HRV analysis specifically designed for adult pain monitoring. This scoring system assigns values between 0 (high pain) and 100 (low pain), with scores generally above 50 indicating adequate pain relief (*Boselli & Jeanne, 2014*). Studies suggest ANI's potential to predict postoperative pain (*Boselli et al., 2014*), guide opioid administration during surgery (*Tribuddharat et al., 2021*), and anticipate the need for pain medication after surgery (*Turan et al., 2017*), making it a valuable tool in pain management.

Expanding on the success of ANI in adults, researchers developed the Newborn Infant Parasympathetic Evaluation (NIPE[TM], MDoloris Medical Systems, Loos, France) specifically for children under 2 years. NIPE helps assess pain states in this vulnerable population, including prolonged pain, acute pain, discomfort, and comfortable states (*Butruille et al., 2015*). The NIPE device provides scores ranging from 0 to 100, reflecting the level of parasympathetic nervous system activity. Studies suggest that values below 50 are associated with pain, stress, or discomfort during surgery or other procedures (*Verweij, Kivits & Weber, 2021*; *Lim, 2019*; *Zhang et al., 2019*). While a literature review indicates NIPE's potential as a pain monitoring tool during procedures, research on its use during surgery under general anesthesia remains limited, with mixed findings (*Recher et al., 2021*).

This study aimed to evaluate whether the NIPE could be a more effective tool than relying on clinical signs alone to guide the administration of fentanyl during surgery in children under 2 years. The researchers hypothesized that using NIPE to guide fentanyl administration would lead to optimal pain control during surgery, resulting in less postoperative pain for the children.

## MATERIALS AND METHODS

The study protocol received approval from the Khon Kaen University Ethics Committee in Human Research (HE651243) on July 21, 2022, adhering to the principles outlined in the Declaration of Helsinki and the ICH Good Clinical Practice guidelines. The study was registered with ClinicalTrials.gov (NCT05758090) on February 23, 2023. Prior to enrollment, all patients provided written informed consent by their parents or guardians. The results were reported following the CONSORT (Consolidated Standards of Reporting Trials) guidelines.

This study utilized a prospective, randomized controlled design with 35 participants in each group. The sample size was calculated based on a previous study (*Benchetrit et al., 2021*) reporting an average postoperative pain score of 4.0 on the first day (day 0) with a pooled standard deviation of 2.14 in children following ear surgery. We aimed to detect a clinically significant difference of 1.5 points in pain scores with 80% power and a 5% significance level (alpha = 0.05), accounting for a potential 10% dropout rate. Randomization was conducted using computer-generated blocks of four in a 1:1 ratio. Allocation sequences were kept concealed in sealed envelopes until the time of enrollment.

Eligible participants included children aged up to 2 years, with an American Society of Anesthesiologists (ASA) physical status classification of I or II, undergoing elective surgery on the head, neck, or upper extremities at Srinagarind Hospital, Khon Kaen, Thailand. Children with cardiac arrhythmias, prematurity, planned combined regional block anesthesia, anticipated postoperative intensive care unit (ICU) admission, or those whose parents or guardians declined participation were excluded from the study.

Following randomization into either group N (NIPE) or group C (Control), patients were monitored using standard equipment including electrocardiogram, pulse oximeter, non-invasive blood pressure, capnography, temperature, minimum alveolar concentration (MAC), and NIPE. In group C, the NIPE device was concealed with an opaque cloth to

prevent the attending anesthesiologist from viewing the NIPE data throughout the surgery. After anesthesia completion, NIPE data was downloaded for later analysis. Importantly, the patients and outcome assessors remained blinded to the group allocation throughout the study.

The primary outcome was postoperative FLACC scores in the post-anesthesia care unit (PACU) at 0, 30, 60, and 120 min. Secondary outcomes included intraoperative and postoperative fentanyl consumption, postoperative sedation levels, and side effects such as postoperative respiratory depression and PONV in the PACU. We also investigated the potential correlation between NIPE values and both heart rate and blood pressure.

All patients received standardized anesthesia following institutional guidelines. Patients fasted for solids for 6 h, however, they were allowed clear fluids up to 2 h before surgery. In the operating room, if an intravenous (IV) line was already established, induction commenced with intravenous medications: fentanyl 1–2 µg/kg followed by propofol 2 mg/kg. For patients without an IV line, inhalation induction with sevoflurane was initiated, followed by IV line placement and fentanyl administration at the same dosage. Intubation was facilitated with succinylcholine 1.5–2 mg/kg IV. After successful intubation, all patients received cisatracurium 0.15–0.2 mg/kg IV and were mechanically ventilated with a mixture of oxygen and nitrous oxide (0.5:0.5 L/min) combined with sevoflurane or desflurane to maintain anesthesia depth at 1 MAC. Dexamethasone and ondansetron 0.1–0.2 mg/kg were administered for PONV prophylaxis.

During surgery, fentanyl administration in group N was guided by NIPE scores, aiming to maintain a range between 50 and 70. When the NIPE score dropped below 50, fentanyl 0.5 µg/kg was administered intravenously. In group C, fentanyl 0.5 µg/kg was given intravenously upon a 20% increase in heart rate or mean arterial pressure (MAP). Cisatracurium 0.04–0.05 mg/kg was administered intravenously every 45 min in both groups. Following surgery, reversal of neuromuscular blockade was achieved with atropine 0.02 mg/kg and neostigmine 0.05 mg/kg. Extubation was performed when patients met pre-established normal consciousness, respiration, and motor function criteria. All patients were transferred to the PACU for postoperative monitoring for 2 h.

In the PACU, continuous monitoring included pain assessment using the FLACC score (Facial expression, Legs, Activity, Cry, Consolability; each scored 0–2, resulting in a total score of 0–10) at 0, 30, 60, and 120 min postoperatively. If the FLACC score exceeded 3, fentanyl 0.5–1 µg/kg was administered intravenously, and the total fentanyl dose in the PACU was recorded. Additionally, sedation scores (0 = awake; 1 = easily roused; 2 = roused with difficulty; 3 = difficult to rouse), PONV, and respiratory depression signs (apnea, chest retractions, grunting, cyanosis, lethargy, or decreased oxygen saturation) were documented at the same time points. All data were analyzed using an intention-to-treat approach.

## Statistical analyses

Data normality was assessed using the Shapiro-Wilk test. Normally distributed data are presented as mean ± standard deviation (SD), while non-normally distributed data are

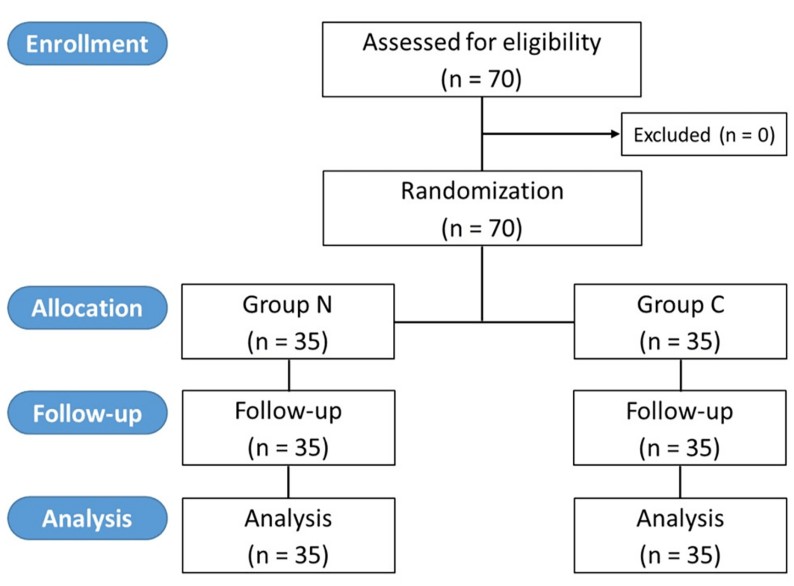

**Figure 1 CONSORT diagram of the study.**

presented as median and interquartile range (IQR). Categorical data are presented as frequencies (%). Group differences were analyzed using appropriate tests: unpaired *t*-test for normally distributed continuous data, Mann-Whitney *U* test for non-normally distributed data, chi-square test for categorical data, and repeated measures ANOVA for between-within-subject comparisons over time. Pearson correlation analysis was used to assess correlations. A *p*-value less than 0.05 was considered statistically significant. All statistical analyses were performed using SPSS 16.0 for Windows (SPSS, Chicago, IL, USA).

## RESULTS

From March to December 2023, 70 patients with 35 in each group were recruited and analyzed (Fig. 1). There were no dropouts. All patients underwent inhalation induction with sevoflurane. The demographic and clinical data of both groups were comparable (Table 1).

The postoperative FLACC scores at 0, 30, 60, and 120 min in the PACU of both groups were similar ($p = 0.446$). The intraoperative, and postoperative fentanyl consumption and sedation scores were comparable between both groups ($p = 0.629, 0.688$, and $0.842$) (Table 2). Additionally, there were no reported cases of postoperative respiratory depression or PONV in either group.

The intraoperative NIPE scores of group N are slightly lower, without statistical significance than group C ($p = 0.620$). However, the NIPE scores of both groups are higher than 50 (approximately 55–60) (Fig. 2). The intraoperative heart rate and MAP of both groups are comparable ($p = 0.703$ and $0.485$) (Figs. 3 and 4).

There are strong correlations between both groups in terms of NIPE scores ($r = 0.735$, $p < 0.0001$), heart rate ($r = 0.630$, $p < 0.0001$), and MAP ($r = 0.846$, $p < 0.0001$) (Fig. 5). In

**Table 1 Demographic and clinical data of the patients (*n* = 70).**

|  | Group C<br>*n* = 35 | Group N<br>*n* = 35 | *p* value |
|---|---|---|---|
| Sex (male) | 17 (48.6) | 22 (62.9) | 0.336 |
| Age (month) | 10.6 ± 6.6 | 11.4 ± 9.6 | 0.685 |
| Weight (kg) | 8.2 ± 2.3 | 7.7 ± 2.2 | 0.356 |
| Height (cm) | 70.7 ± 9.5 | 67.1 ± 9.3 | 0.111 |
| Operation time (min) | 164.1 ± 52.0 | 177.0 ± 52.4 | 0.306 |
| Blood loss (mL) | 5 (2.0–20.0) | 10 (5.0–50.0) | 0.061 |
| Diagnosis: |  |  | 0.358 |
| Bilateral otitis media | 2 (5.7) | 1 (2.9) |  |
| Cataract | 1 (2.9) | 1 (2.9) |  |
| Congenital esotropia | 1 (2.9) | 0 (0) |  |
| Ptosis | 1 (2.9) | 0 (0) |  |
| Unilateral cleft lip | 11 (31.4) | 14 (40.0) |  |
| Unilateral cleft palate | 0 (0) | 3 (8.5) |  |
| Unilateral cleft lip & palate | 19 (54.2) | 16 (45.7) |  |

**Note:**
Data are presented as *n* (%), mean ± SD, or median (interquartile range).

**Table 2 Primary and secondary outcomes (*n* = 70).**

|  | Group C<br>*n* = 35 | Group N<br>*n* = 35 | *p* value |
|---|---|---|---|
| Postoperative FLACC score at: |  |  | 0.446 |
| 0 min | 7 (5–8) | 7 (3–8) |  |
| 30 min | 2 (0–5) | 2 (0–5) |  |
| 60 min | 1 (0–2) | 0 (0–2) |  |
| 120 min | 0 (0–1) | 0 (0–0) |  |
| Intraoperative fentanyl consumption (µg) | 21.9 ± 10.9 | 23.4 ± 15.3 | 0.629 |
| Postoperative fentanyl consumption (µg) at: |  |  | 0.688 |
| 0 min | 5 (0–5) | 4 (0–5) |  |
| 30 min | 0 (0–3) | 0 (0–5) |  |
| 60 min | 0 (0–0) | 0 (0–0) |  |
| 120 min | 0 (0–0) | 0 (0–0) |  |
| Postoperative sedation score at: |  |  | 0.842 |
| 0 min | 0 (0–0) | 0 (0–0) |  |
| 30 min | 1 (0–1) | 1 (0–1) |  |
| 60 min | 1 (0–1) | 0 (0–1) |  |
| 120 min | 0 (0–0) | 0 (0–0) |  |

**Note:**
Data are presented as mean ± SD, or median (interquartile range).

both group C and group N, the NIPE scores reveal strong negative correlations with heart rate ($r = -0.610$, $p < 0.0001$, and $r = -0.674$, $p < 0.0001$) (Fig. 6) and MAP ($r = -0.860$, $p < 0.0001$, and $r = -0.756$, $p < 0.0001$) (Fig. 7).

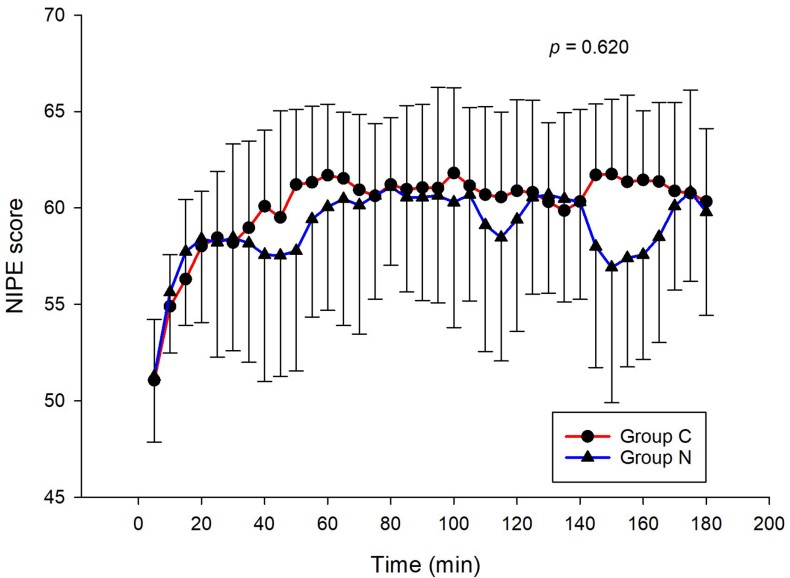

**Figure 2 Comparison of intraoperative NIPE scores between group C and group N.**

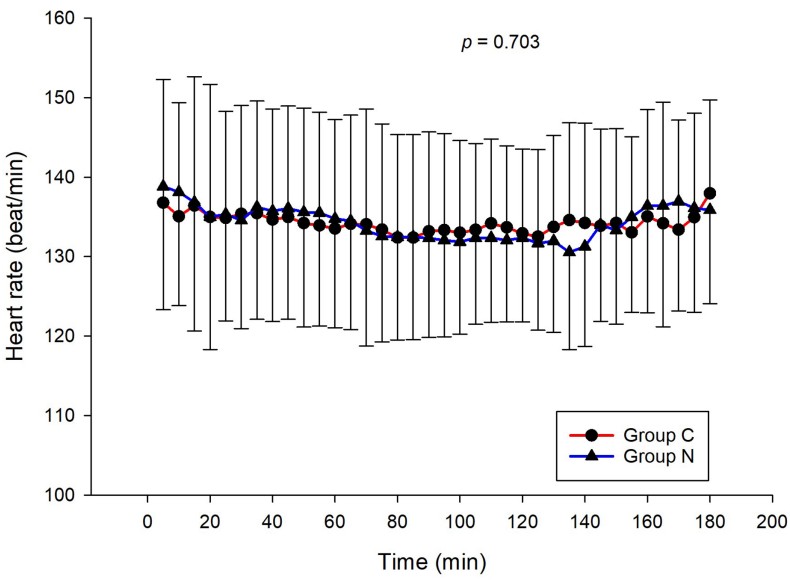

**Figure 3 Comparison of intraoperative heart rates between group C and group N.**

## DISCUSSION

The results of the current study show that intraoperative fentanyl administration guided by the NIPE score or vital signs (heart rate or MAP) yields similar outcomes in terms of postoperative FLACC scores, intraoperative and postoperative fentanyl consumption, and sedation scores. Intraoperative fentanyl administration guided by NIPE score is as accurate as guided by heart rate or MAP, reflected by similar NIPE score, heart rate, and MAP between both groups (Figs. 2–4). The consistently, lower NIPE scores, although without

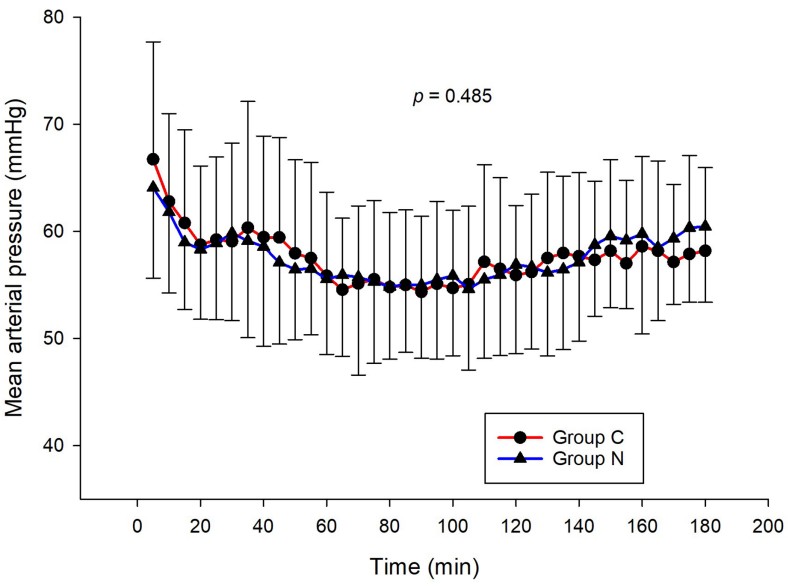

**Figure 4 Comparison of mean arterial pressures between group C and group N.**

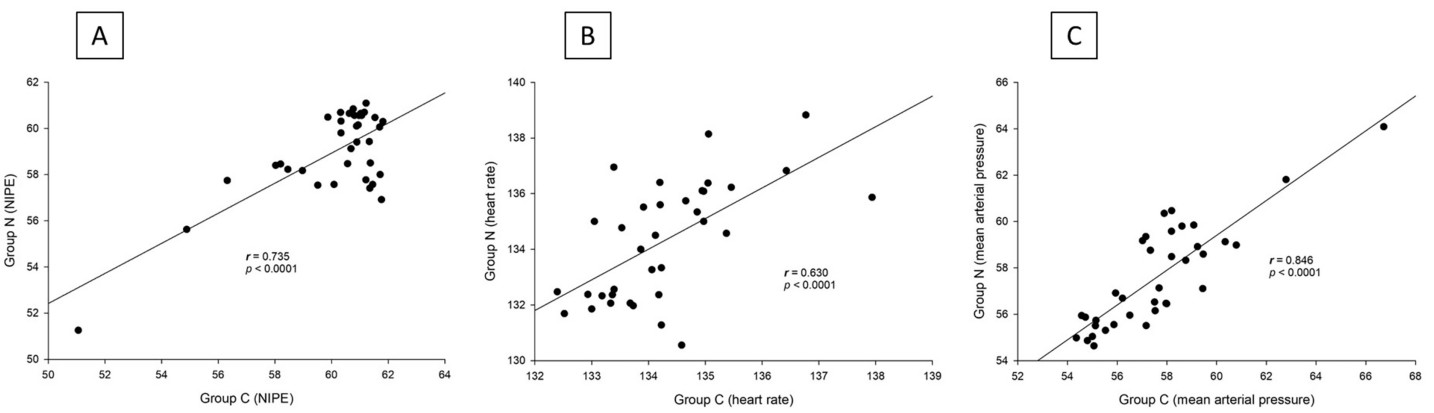

**Figure 5 (A–C) Correlation analysis: (A) NIPE scores, (B) Heart rates, and (C) Mean arterial pressures between group C and group N.**

significant difference, in group N compared to the group C (Fig. 2) might suggest that NIPE-guided fentanyl administration resulted in a more targeted approach, potentially minimizing the need for excessive fentanyl use. Strong correlations between NIPE, heart rate, and MAP in both groups (Fig. 5) suggest that intraoperative fentanyl administration guided by NIPE leads to similar outcomes as when guided by changes in vital signs. The low FLACC score reflects the optimal dose of fentanyl in both groups. Furthermore, NIPE scores show strong negative correlations with both heart rate and MAP in both groups (Figs. 6 and 7), which reflect the validity of NIPE as a tool to monitor nociception. High nociceptive pain leads to low NIPE scores and high heart rate or MAP, and vice versa.

Our findings partially align with prior research. *Neumann et al. (2022)* found a correlation between NIPE and pain stimuli (venous puncture, intubation, skin incision) in

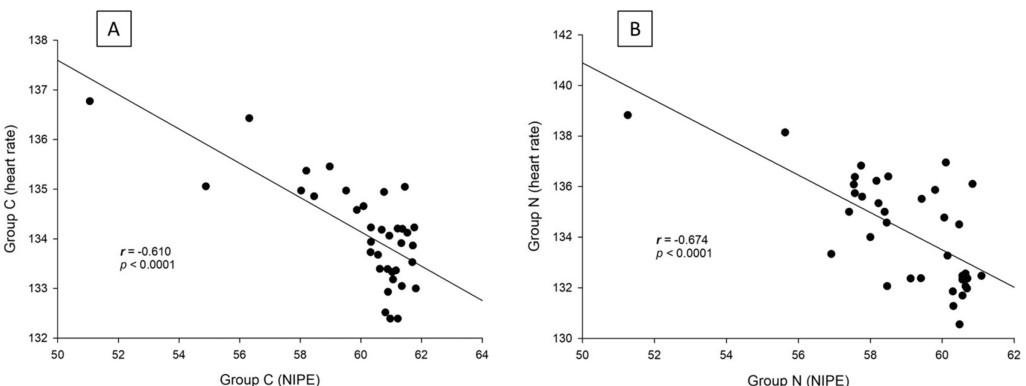

**Figure 6 Correlation analysis within groups: (A) NIPE scores and heart rates in group C; (B) NIPE scores and heart rates in group N.**

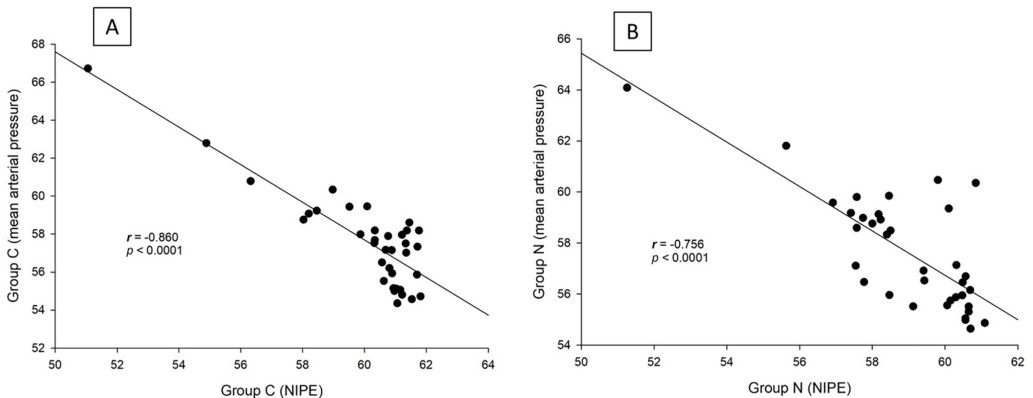

**Figure 7 Correlation analysis within groups: (A) NIPE scores and mean arterial pressures in group C; (B) NIPE scores and mean arterial pressures in group N.**

51 children under 2 years during surgery. However, they did not observe changes in heart rate. This difference might be due to the milder and more intermittent nature of the stimuli used in their study compared to the continuous and intense stimulation in our study. Similarly *Weber et al. (2019)*, reported increased NIPE values after additional opioid administration in 67 infants undergoing procedures (aged 0–2 years) with NIPE scores below 50. However, they observed no heart rate changes. A possible explanation could be the use of caudal blocks in 37.3% of their patients, a pain management technique not employed in our study.

The strong negative correlations of NIPE with both heart rate and MAP which reflect pain response in this study are similar to many studies. In a prospective observational study in neonates and infants undergoing elective day-case surgery, *Ivanic et al. (2023)* demonstrated that intraoperative NIPE indices had a significant negative correlation with immediate postoperative FLACC scores ($r = -0.31$, $p = 0.03$). *Recher et al. (2020)* evaluated the NIPE index and Comfort Behavior (COMFORT-B) scores during care procedures in 32 children under 3 years old in a pediatric intensive care unit (PICU). They found a

significant inverse correlation between the NIPE index and COMFORT-B scores ($r = -0.44$, $p = 0.0001$). They concluded that the NIPE index was valid for assessing distress in children in PICU.

There are many studies assessing the ability of the NIPE index to detect acute procedural pain in newborn infants with varying results. In a descriptive study of 121 children under 2 years undergoing general anesthesia with 1,222 observations of FLACC and COMFORT-B scores by *Verweij, Kivits & Weber (2021)*, they found that the NIPE detected pain and discomfort after general anesthesia with high area under the ROC curve (0.77 for FLACC $\geq 4$, 0.81 for COMFORT-B $\geq 17$, and 0.77 for FLACC $\geq 4$ & COMFORT-B $\geq 17$). A recent systematic review by *Sakthivel et al. (2024)* analyzing 10 non-randomized studies with 548 participants found NIPE promising for detecting both intraoperative and early postoperative pain, suggesting its potential usefulness in pain assessment during procedures. However, the studies had varied results, highlighting the need for further research to confirm its effectiveness.

Unlike previous descriptive studies, ours employed a randomized design to directly compare the effectiveness of NIPE with heart rate or MAP in guiding fentanyl administration during surgery for children under 2 years. We evaluated various parameters, including postoperative FLACC scores, fentanyl use, sedation levels, NIPE scores, heart rate, and MAP. Interestingly, the strong correlations observed between NIPE, heart rate, and MAP in both groups (Fig. 5) suggest that both methods might be equally effective for guiding intraoperative fentanyl administration.

Furthermore, NIPE scores in both groups remained above 50 (Fig. 2), indicating adequate pain control. Additionally, the strong negative correlations between NIPE and heart rate (Fig. 6), and MAP (Fig. 7) further support the validity of NIPE as a tool for assessing pain in children under 2 years.

Traditional methods relying on a 20% increase in heart rate or MAP to guide fentanyl administration in newborns and infants pose challenges. Baseline vital signs vary significantly in this age group, making it difficult to detect a definitive threshold for additional medication. NIPE scores, on the other hand, offer a more trackable and potentially more accurate way to guide fentanyl dosing due to their focus on parasympathetic activity.

This study has some key advantages. The randomized design helps ensure the results are reliable by minimizing the influence of other factors. Additionally, we evaluated NIPE's effectiveness using a variety of measures, which strengthens the validity of our findings. However, this study has limitations. First, it was conducted at a single center with a limited number of participants. Second, the surgical procedures involved resulted in mild to moderate pain, potentially limiting the generalizability of the findings to other hospitals or more severe types of surgery. To address these limitations and enhance generalizability, future research should involve a larger, multicenter sample and consider including procedures with a wider range of pain severity. This study used succinylcholine to facilitate endotracheal intubation. While it is not absolutely contraindicated in infants, it can cause serious, though rare, side effects, particularly malignant hyperthermia. Therefore, its use should be avoided when safer alternatives are available.

## CONCLUSION

NIPE-guided fentanyl administration was not superior to heart rate/MAP-guided administration, as both approaches yielded similar outcomes in terms of postoperative pain, fentanyl consumption, sedation, and intraoperative measurements. However, NIPE may offer a more practical and precise method for guiding fentanyl administration in children under 2 years, as it is an objective tool with a defined threshold. This could lead to more targeted pain management. While this study highlights NIPE's potential as a valuable tool for pain management in young children undergoing surgery, further research with larger, multicenter studies is needed to confirm its effectiveness for widespread use.

### Funding

The authors received no funding for this work.

### Competing Interests

The authors declare that they have no competing interests.

### Author Contributions

- Darunee Sripadungkul conceived and designed the experiments, performed the experiments, analyzed the data, prepared figures and/or tables, authored or reviewed drafts of the article, and approved the final draft.
- Sirirat Tribuddharat conceived and designed the experiments, performed the experiments, analyzed the data, prepared figures and/or tables, authored or reviewed drafts of the article, and approved the final draft.
- Thepakorn Sathitkarnmanee conceived and designed the experiments, performed the experiments, analyzed the data, prepared figures and/or tables, authored or reviewed drafts of the article, and approved the final draft.
- Pimprapa Muenpirom conceived and designed the experiments, performed the experiments, analyzed the data, prepared figures and/or tables, authored or reviewed drafts of the article, and approved the final draft.
- Panaratana Ratanasuwan conceived and designed the experiments, performed the experiments, analyzed the data, prepared figures and/or tables, authored or reviewed drafts of the article, and approved the final draft.

### Human Ethics

The following information was supplied relating to ethical approvals (*i.e.*, approving body and any reference numbers):

The Khon Kaen University Ethics Committee in Human Research (HE651243).

### Clinical Trial Ethics

The following information was supplied relating to ethical approvals (*i.e.*, approving body and any reference numbers):

The Khon Kaen University Ethics Committee in Human Research (HE651243).

## Data Availability

The raw measurements are available in the Supplemental Files.

## Clinical Trial Registration

The following information was supplied regarding Clinical Trial registration:

NCT05758090

## Supplemental Information

Supplemental information for this article can be found online at http://dx.doi.org/10.7717/peerj.18267#supplemental-information.

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
