# Peer review of "Effectiveness of newborn infant parasympathetic evaluation (NIPE) index for guiding intraoperative fentanyl administration in children under 2 years: a randomized controlled trial"

_PeerJ, doi:10.7717/peerj.18267_

## Round 0.1 · original submission · Major Revisions

Please address the concerns of both reviewers. Note the comments of R1 that either more data, and/or more explicit discussion of the limitations is needed.

·

Basic reporting

This study is a very interesting randomized controlled trial which compared the effectiveness of NIPE-guided fentanyl administration to traditional vital signs (heart rate and mean arterial pressure) in managing pain during surgery in children under 2 years.
However, currently, some issues require appropriate answers and corrections.

First, this study established the following objectives and hypotheses: lines 89-92
This study aimed to evaluate whether the NIPE could be a more effective tool than relying on clinical signs alone to guide the administration of fentanyl during surgery in children under 2 years. The researchers hypothesized that using NIPE to guide fentanyl administration would lead to optimal pain control during surgery, resulting in less postoperative pain for the children.
Nonetheless, this study showed that both NIPE-guided and heart rate/MAP-guided intraoperative fentanyl administration achieved similar pain management outcomes in this study. Therefore, since this study is not a non-inferiority clinical trial, it is judged that the conclusion should be more focused on the fact that NIPE was not better than hemodynamic guided analgesia. In other words, it should be emphasized that NIPE did not provide better analgesia compared to hemodynamics in infant patients, so its use is limited.

Experimental design

As a next issue, although this study is an RCT, there are parts where the anesthesia protocol is not well refined. A more unified and uniform protocol should have been applied in the following areas (lines 127-136), and these points should be specified in the limitations. In particular, the use of succinylcholine in infant patients appears to be very unfavorable. Succinylcholine is a drug that is contraindicated in infant patients because it has many side effects including even malignant hyperthermia most severely. In particular, it is considered a drug that should have been excluded from the NIPE study because it can affect the autonomic nervous system. Also, anesthesia methods should have been applied in a unified manner rather than including two or more methods as options. For example, the effects of inhalation induction and iv induction on the autonomic nervous system will clearly be different. Please clearly indicate in the table how many patients in the two groups applied these two methods. The same applies to the use of sevoflurane or desflurane. I am personally opposed to using desflurane in infant patients.

Validity of the findings

no comment

Additional comments

Overall, the subject matter and design of this study are good and demonstrate clinically interesting results.

·

Basic reporting

The language is clear and understandable
It is a well written article and with clarity in communicating the hypothesis and the details of the study conducted. The references need to be put in the Vancouver format. The table 1 is incomplete with respect to the p values. In table 2- p value is not mentioned again for all the columns. The difference between the two groups at the separate time intervals will be denoted as separate p values. Kindly review and make the changes.

Experimental design

No comment

Validity of the findings

No comment

Additional comments

The authors have drafted a good article. Can be acceptable with above revisions.

---

## Round 0.2 · accepted · Accept

Thank you for making the changes that the reviewers requested in their comments.

·

Basic reporting

This study is a very interesting randomized controlled trial which compared the effectiveness of NIPE-guided fentanyl administration to traditional vital signs (heart rate and mean arterial pressure) in managing pain during surgery in children under 2 years.
In addition, currently, I confirm some major issues require appropriate answers and corrections were properly resoluted by the authors.

Experimental design

I confirm some major issues associated with experimental design were properly resoluted by the authors.

Validity of the findings

no comment

Additional comments

Overall, the subject matter and design of this study are good and demonstrate clinically interesting results. Some major issues associated with experimental design were properly resoluted by the authors.